# Lasting effects of the COVID-19 pandemic on language processing

**Daniel Kleinman**[1]☯*, **Adam M. Morgan**[2]☯, **Rachel Ostrand**[3]☯, **Eva Wittenberg**[4]☯

**1** Haskins Laboratories, New Haven, CT, United States of America, **2** NYU School of Medicine, New York, NY, United States of America, **3** IBM Research, Yorktown Heights, NY, United States of America, **4** Department of Cognitive Science, Central European University, Vienna, Austria

☯ These authors contributed equally to this work.
* daniel.kleinman@yale.edu

**Data Availability Statement:** The stimuli, materials, deidentified trial-level data, and analysis code for all experiments are publicly accessible on the Open Science Framework repository at https://osf.io/dxc97/ (DOI 10.17605/OSF.IO/DXC97).

## Abstract

A central question in understanding human language is how people store, access, and comprehend words. The ongoing COVID-19 pandemic presented a natural experiment to investigate whether language comprehension can be changed in a lasting way by external experiences. We leveraged the sudden increase in the frequency of certain words (*mask*, *isolation*, *lockdown*) to investigate the effects of rapid contextual changes on word comprehension, measured over 10 months within the first year of the pandemic. Using the phonemic restoration paradigm, in which listeners are presented with ambiguous auditory input and report which word they hear, we conducted four online experiments with adult participants across the United States (combined *N* = 899). We find that the pandemic has reshaped language processing for the long term, changing how listeners process speech and what they expect from ambiguous input. These results show that abrupt changes in linguistic exposure can cause enduring changes to the language system.

## Introduction

Most cognitive research on language processing relies on laboratory-based experiments. While this has proven a fruitful avenue for theory-building, it is critical to validate these theories in ecologically valid settings. However, the opportunities for this are few and far between: Capturing and experimentally measuring cognitive effects in real-time situations, particularly across a large sample of participants, presents unique challenges.

Some researchers have found creative ways of achieving this. For instance, Brown and Kulik pioneered research into flashbulb memories by asking people when and how they learned about a major, emotionally charged event shortly after the event occurred (e.g., the assassination of John F. Kennedy, the fall of the Berlin Wall, or the 9/11 terrorist attacks) [1]. In other flashbulb memory experiments, participants were retested months or years later to investigate the reliability and consistency of their memories [2, 3]. Studies that leverage the unique properties of these events to shed light on underlying cognitive phenomena can make valuable contributions to the field. However, major events such as these occur only rarely and unpredictably, so opportunities to address cognitive questions in a real-world situation are few and far between.

**Funding:** This research was funded by a UC San Diego Innovation Grant of Inclusive Research Excellence (to EW; https://diversity.ucsd.edu/centers-resources/funding.html) and an IBM Global University Award Faculty Grant (to EW; https://research.ibm.com/university/awards/shared_university.html). RO is an employee of, and receives salary from, IBM. The funders had no role in study design, data collection and analysis, decision to publish, or preparation of the manuscript.

**Competing interests:** The authors have declared that no competing interests exist.

One such major event is the COVID-19 global pandemic, which has altered society on an unprecedented scale. Among the many changes it has wrought is a shift in public discourse, and consequent changes to people's daily vocabulary: Words like *mask*, *isolation*, and *lockdown* became much more common practically overnight [4]. This presented (what we hope is) a once-in-a-generation opportunity to study rapid changes in language processing *in situ*. In the present research, we leveraged the sudden, massive increase in the frequency of these words to investigate the effects of rapid contextual changes on language processing, their dynamic development over time, and whether linguistic access is mediated by non-linguistic external factors.

## Effects of the pandemic on language processing

The cognitive representation of words is characterized by two main features. First, words are stored and accessed depending on how often and in what contexts they appear. The sensitivity of language users to the statistics of their linguistic input makes them faster both to produce and to comprehend common words compared to uncommon ones [5, 6]; such *frequency effects* are among the most well-documented in cognitive science. Second, lexical knowledge is bound to semantic knowledge. This semantic linkage gives rise to a well-attested effect in psycholinguistics, *semantic priming*, in which comprehending an initial word (*dog*) speeds recognition of an associatively-related, subsequently presented word [*cat*; 7]. Remarkably, such priming effects also hold for non-linguistic sounds with strong semantic associations: The sound of a cat's meow also speeds recognition of the word *cat* [8, 9].

From the very first days of the pandemic, people's daily vocabulary underwent a drastic shift, affecting lexical processing on both of these dimensions–statistical and semantic. Formerly uncommon words like *mask* and *lockdown* (and their translation equivalents in languages around the world) experienced a sudden, substantial increase in frequency for billions of people, as the social discourse abruptly shifted for entire nations practically simultaneously [4]. At the same time, concepts like *coughing* and *isolation*, which had previously shared at best a minimal association, became tightly linked nodes in a novel semantic network.

These changes presented a unique opportunity to measure how the language processing system adapts to and learns from novel input in an entirely natural way. By measuring changes caused by the societal effects of the COVID-19 pandemic to the language processing system, we traced a naturally-induced, population-wide shift in how humans store and access words.

## Effects of experience on lexical knowledge

It has long been known that lexical knowledge can be altered by recent experience. For instance, exposure to an uncommon word (e.g., a low-frequency word such as *anvil*) or the non-dominant meaning of a homonym (e.g., *ball*, meaning a fancy party rather than sports equipment) facilitates subsequent comprehension of that word or meaning by increasing its accessibility [10, 11]. However, these effects are transient, returning to pre-exposure levels in a matter of minutes, hours or days. For instance, in collaboration with the BBC, Rodd and colleagues ran a large-scale experiment in which they aired a radio program which used ambiguous homophonous words in disambiguating contexts, such as "The princess wore a beautiful gown to the *ball*." [12]. After the radio program, listeners participated in a survey and provided associations to the ambiguous homophones they had heard on the program. Participants were more likely to generate words related to the primed, subordinate meaning, compared to participants who had not heard the program. However, this effect lasted for only a few hours after the program aired. This transience suggests that such demonstrations result from paradigm-

or context-specific adaptations, rather than reflecting lasting changes to the language system more globally.

One limitation of prior research in this area is that it has generally measured short-term adaptations induced by a circumscribed or unnaturally constrained linguistic context, such as a single experimental session. In contrast, the pandemic, by precipitating global changes to both the statistical and semantic properties of words, allowed us to test for corresponding changes to word knowledge resulting from an ecologically valid manipulation, and to do so over the long term.

### The present study

A crucial component which differentiates the present experiments from prior work [e.g., 12] is that we do not *manipulate* listeners' recent linguistic input in order to measure the cognitive outcome; instead, our experiments measure the real-world cognitive effects of an external shift that profoundly affected people's lives. That is, the present experiments use the *external world* as a prime and measure how our brains have learned and stored words as a result of lived experience, investigating changes to the language system that are the result of natural changes in the linguistic environment.

With these experiments, we address three questions: (a) Did the sudden increase in frequency of certain words alter the *statistical* aspect of lexical representation? Specifically, did newly frequent words like *mask* become more readily comprehensible and accessible? (b) Did the pandemic create a novel semantic network, altering the *semantic* aspect of lexical representation? Specifically, is the non-linguistic sound of coughing now semantically linked to pandemic-related words like *mask*, so that the sound facilitates lexical comprehension of such words? (c) Can naturally-induced changes in lexical representation persist beyond the duration observed in the lab? Specifically, did these cognitive changes persist over 10 months?

### Open practices statement

The stimuli, materials, deidentified trial-level data, and analysis code for all experiments are publicly accessible at https://osf.io/dxc97/. The study design, stimuli list, participant recruitment criteria, sample size, data exclusion criteria, and statistical analysis plan for all experiments after the first were preregistered (*Exp. 2a*: https://osf.io/kp6ac/; *1b*: https://osf.io/n6e8g/; *2b*: https://osf.io/3rwsx/). Preregistrations specified the model-pruning approach, one-tailed hypothesis tests and exclusion criteria for participants and items, and were followed precisely in all respects except that sample sizes for two experiments differed by 1–2% from planned *N*s due to a technical error (see S1 Text section *Method*: *Deviations from Preregistrations* for details). Analyses for Experiment 2a were preregistered prior to data collection, and analyses for Experiments 1b and 2b were preregistered after data collection but prior to analysis (though condition-blinded data was used to determine exclusion criteria, as described in detail in the preregistrations and in S1 Text section *Changes in recruitment and criterion-setting to improve data retention*).

### Materials and methods

We conducted four experiments over ten months (April 2020 –February 2021) during the first, second, and third waves of the COVID-19 pandemic in the United States (*Exp. 1a*: April 2020; *Exp. 2a*: July 2020, *Exp. 1b* [replication of *1a*]: February 2021, *Exp. 2b* [replication of *2a*]: February 2021). As many methodological details were shared between experiments, we report them jointly below.

## Participants

Participants were recruited and tested via Amazon Mechanical Turk. Individuals were eligible to participate only if they were a (self-reported) native English speaker with an IP address located in the United States, and had an approved HIT (task) rate of at least 98% (Experiments 1a and 2a) or 90% (Experiments 1b and 2b). Participants in Experiments 1b and 2b were further screened using CloudResearch based on the participants' study completion history to improve data quality: They were on the CloudResearch Approved List and had previously completed between 500 and 5000 HITs on the platform (vs. no minimum number of HITs required for Experiments 1a and 2a, an oversight that yielded a large number of participants with no prior HITs and thus poor data quality). Participants gave written informed consent; ethical approval was provided by the University of California, San Diego Institutional Review Board. The experiment took approximately 10 minutes to complete and participants were compensated with $2.50.

The total number of participants recruited for each experiment, as well as the number who contributed data that were ultimately included in analyses, are shown in the in Table A in S1 Text. Across experiments, 899 participants contributed usable data.

In the first experiment (Exp. 1a), we set an a priori target *N* of 240 participants–nearly an order of magnitude more than in prior phonemic restoration experiments. We chose this high number because several features of our experimental design made it difficult to estimate effect size (e.g., our use of novel stimuli, a relatively low trial count, running the paradigm in people's homes instead of a carefully-controlled auditory environment and equipment in the lab). Two of the three effects that we expected to observe in Experiment 1a reached statistical significance with that sample size. To increase statistical power in the next experiment (Exp. 2a), we doubled the number of participants and increased the number of stimuli from 16 to 20. Poor data quality in Experiment 2a led us to improve our participant screening measures for Experiments 1b and 2b, for which we returned to our originally-targeted sample size of 240 participants. (Note that the sample sizes for all experiments after Experiment 1a were preregistered.)

## Experimental paradigm

Experiments used the *phonemic restoration* experimental paradigm [13], in which an audio recording of a spoken word is altered by removing a short segment and replacing it with noise. Listeners report "hearing" sounds that are not actually present in the acoustic signal. This cognitive "restoration" process allows listeners to understand speech even when the acoustic signal is noisy, as often occurs in real-world environments. Phonemic restoration combines the bottom-up information of the raw acoustic signal with top-down lexical expectations, so that when hearing "abra#adabra" (# indicates a non-speech noise), listeners restore the missing /k/ sound and report hearing *abracadabra*, but not nonexistent words like *abra-ta-dabra*. The phonemic restoration induced by a recording is a function of the listener's linguistic expectations formed from the immediate lexical or semantic context. When multiple real words are consistent with the acoustic input (e.g., /#æsk/ can be restored to *mask* or *task*, among others), top-down knowledge–which can take the form of semantic context, or expectations drawn from linguistic experience–biases the restoration toward a relevant word. Stronger context–for example, as induced by higher frequency (and thus more globally expected) words, or longer words which have more phonetic information–induces greater lexical expectations in listeners and thus greater phonemic restoration [14].

## Materials and design

Experimental stimuli were 16 (Exps. 1a & 1b) or 20 (Exps. 2a & 2b) audio recordings of words which had 1–2 phonemes removed and replaced by noise. (The only experimental design

difference between Experiments 1 and 2 was the specific stimuli used.) On each trial, participants heard a recording and typed the word they thought they heard. Each ambiguous stimulus could be restored to (at least) two real words; for example, /#æsk/ could be restored to either *mask* (whose frequency increased due to COVID: TARGET response) or *task* (frequency unaffected by COVID: COMPETITOR response). We measured how often listeners reported hearing the TARGET as compared to the COMPETITOR word: If the sudden increase in frequency of pandemic-related words impacted how easily these words are accessed, then listeners presented with an ambiguous auditory stimulus should perceive the TARGET word more often than the COMPETITOR word.

The stimuli were designed in quadruplets: a COVID vs. CONTROL manipulation, each with a TARGET and a COMPETITOR response option (see Fig 1A). The COVID-TARGET

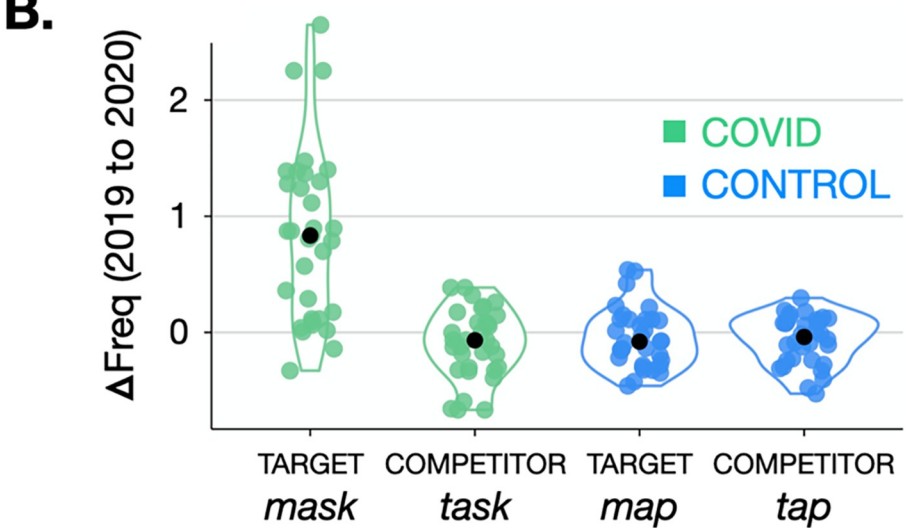

**Fig 1. Visualizations of critical stimulus properties.** (A) Example stimulus quadruplet. For each COVID/CONTROL pair, the COMPETITOR word was recorded, and the critical sound was removed and replaced with a COUGH or NOISE. Participants' responses were coded as TARGET or COMPETITOR if they were one of the two pre-defined response options for each stimulus, or otherwise excluded. (B) Changes in stimulus (log) word frequency from 2019 to 2020 by response category. COVID-TARGET words' frequency increased substantially after the onset of the pandemic ($t(96) = 11.96$, $p < .001$), while the frequencies of other response options remained relatively stable (all $|t| < 1.45$, all $p > .15$).

response option was pandemic-related (*mask*); the COVID-COMPETITOR response option was not (*task*). The CONTROL-TARGET (*map*) and CONTROL-COMPETITOR (*tap*) were both pandemic-unrelated, and included the same phonological contrast in the same position as the COVID item (*m/t*); this ensured that any differences between phonemes that might affect restoration (e.g., the presence of noise components in fricatives and voiceless stops) would affect the COVID and CONTROL conditions equally. A native speaker of American English (author D.K.) recorded the COMPETITOR words for both COVID and CONTROL items; thus, any coarticulation would bias listeners *toward* the COMPETITOR response and, critically, *away* from the TARGET response. Then, the relevant phonemes were removed from the recording (e.g., *t* in *task* and *tap*) and replaced with a pre-recorded COUGH or grey NOISE of the same duration. Each participant was presented with one of four experimental lists; in each list, stimuli were divided equally across the four experimental conditions (COVID/COUGH, COVID/NOISE, CONTROL/COUGH, CONTROL/NOISE), with each stimulus having two response options (TARGET and COMPETITOR). See Table 1 for the complete list of stimuli.

The process for selecting items was as follows. The COVID-TARGET items were chosen first, as words which suddenly increased in frequency in the national discourse due to the pandemic. Then, the COVID-COMPETITOR items were created as minimal pair words for their respective COVID-TARGET items, by finding words which were one phoneme different, balancing the constraints of matched pre-COVID frequency between the TARGET and COMPETITOR words, and attempting to minimize the size of the phonological neighborhood around the minimal pair phoneme which would ultimately be blocked out by the noise. This matching was done using the phoneme-based dictionary *Sylvia* [15], which queries the CMU Pronouncing Dictionary [16] using regular expressions. Subsequently, CONTROL-TARGET and CONTROL-COMPETITOR words were chosen (using *Sylvia*) to have the same minimal pair distinction as the COVID-TARGET and COVID-COMPETITOR pair, in the same location (initial, medial, final) in the word.

Crucial to our experiment was the assumption that, pre-pandemic, the COVID-TARGET words were equally cognitively accessible as the COVID-COMPETITOR and CONTROL--TARGET words, so that any observed differences in lexical processing at the time of data collection (post-pandemic onset) could be attributed to pandemic-induced *changes* in frequency. To verify this assumption, lexical frequency was calculated for each word over two time periods using the News on the Web (NOW) corpus, which consists of time-stamped, web-based newspapers and magazines, capturing shifting lexical trends in real time [17]. Two subsets of the corpus were created: all documents in the NOW corpus from January–December 2019 (the "pre-pandemic" period) and all documents in the NOW corpus from January–December 2020 (the "post-pandemic" period). Frequency was calculated per million words, and log-transformed for computations. For each time period, a one-way ANOVA was conducted to compare word frequencies between items in each of the four groups.

Comparisons confirmed that stimulus items were matched on *pre-pandemic* lexical frequency ($F(3,32) = 0.88$, $p = .455$), but significantly differed on *post-pandemic* frequency ($F(3,32) = 5.68$, $p = .001$). Specifically, COVID-TARGET words were significantly more frequent than the others after pandemic onset: Contrasts (corrected for multiple comparisons via the Tukey method) revealed that, post-pandemic onset, COVID-TARGET words (*mask*) were significantly higher frequency than COVID-COMPETITOR words (*task*; $p = .011$), CONTROL-TARGET words (*map*; $p = .004$), and CONTROL-COMPETITOR words (*tap*; $p = .005$); no other comparisons were significant (see Fig 1B). We also observed that word frequencies in 2019 vs. 2020 were almost perfectly correlated for COVID-COMPETITOR, CONTROL-TARGET, and CONTROL-COMPETITOR words (all $R^2 > .97$), but were less

**Table 1. List of stimuli used in experiments.**

| COVID-TARGET | COVID-COMPETITOR | CONTROL-TARGET | CONTROL-COMPETITOR | Experiment used |
|---|---|---|---|---|
| bed | bet | mad | mat | 1 |
| cancel | council | can't | count | 1 |
| chill | hill | chair | hair | 1 |
| cough | call | stiff | still | 1 |
| crowd | crown | cloud | clown | 1 |
| death | deck | oath | oak | 1 |
| fever | beaver | feet | beat | 1 |
| hand | land | half | laugh | 1 |
| hoarding | boarding | heaping | beeping | 1 |
| lung | rung | lust | rust | 1 |
| mask | task | map | tap | 1 |
| rest | wrecked | chest | checked | 1 |
| sick | sit | kick | kit | 1 |
| soap | soul | rope | roll | 1, 2 |
| spread | sprayed | tread | trade | 1, 2 |
| washing | watching | lashing | latching | 1, 2 |
| corona | corolla | tenor | teller | 2 |
| isolation | oscillation | idly | oddly | 2 |
| lockdown | knockdown | loading | coding | 2 [†] |
| remote | rewrote | timing | tiring | 2 |
| sickness | thickness | sinking | thinking | 2 |
| curve | curl | swerve | swirl | 2 |
| clinic | cynic | classy | sassy | 2 |
| doctor | proctor | deposition | preposition | 2 |
| screening | screaming | cunning | coming | 2 |
| infection | injection | effect | eject | 2 |
| testing | tempting | attested | attempted | 2 |
| cough | call | safe | sail | 2 [*] |
| death | deck | worth | work | 2 [*] |
| fever | beaver | finder | binder | 2 [*] |
| hoarding | boarding | hauling | balling | 2 [*] |
| mask | task | might | tight | 2 [*] |
| sheltering | sweltering | sharing | swearing | 2 |

*Note*. Each row represents a stimulus quadruplet.

[*] COVID item is the same as in Experiment 1, but CONTROL item was changed for Experiment 2.

[†] Due (in two senses) to a coding error, these COVID and CONTROL items differed on the removed phonemes (/l/ and /n/ vs. /l/ and /k/).

tightly linked for COVID-TARGET words ($R^2$ = .73) due to the latter's frequency increase. This is consistent with the idea that changes in COVID-TARGET word frequency from 2019 to 2020 were sudden and could not have been predicted from pre-pandemic corpus statistics alone.

To test the hypothesis that the pandemic affected the semantic aspect of lexical representation, the sound which replaced the deleted phonemes was either a COUGH (pandemic-related) or grey NOISE (pandemic-unrelated). We expected that the coughing sound would associatively prime pandemic-related semantic networks, biasing participants to hear the TARGET word in the COVID condition, but less so in the CONTROL condition.

## Procedure

As described above, on each trial, participants heard a single recording and then typed the word they thought they heard. After they submitted their response, the next trial started. Recordings could not be replayed, and there was no response deadline. After completing all trials, participants completed a post-experiment questionnaire that collected information about participant demographics; what they thought the experiment was about (Exps. 1b and 2b only); and the extent to which they thought about COVID and took COVID-related preventative measures (masking, isolation, etc.) in their lives.

## Response coding and exclusions

The dependent variable was what kind of word participants reported hearing: the TARGET vs. the COMPETITOR. Accordingly, responses were coded as matching the TARGET (*mask* or *map*) or COMPETITOR (*task* or *tap*) (37.9%); non-matching responses (62.1%) were discarded. Non-matching responses consisted of other competitors that were consistent with the auditory input (e.g., *ask* or *flask*, which were consistent with substituting 0+ phonemes for the noise in the stimulus /#æsk/; 16.0%), input-inconsistent responses (e.g., *mast*, which included conflicting phonemes; 39.2%), and non-words and blank responses (7.0%). These percentages were computed based on all data collected, prior to other exclusions. For more details on how and why we excluded participants and items from analysis, and for data exclusion rates for each experiment, see S1 Text section *Method*: *Data Exclusion*. Note that including input-consistent competitors (*ask*, *flask*) in the analyses did not change the statistical significance of any results in the cross-experiment analysis (see S1 Text section *Analyses that included acceptable alternative responses*).

## Analyses and hypothesis testing

For each individual experiment and for the cross-experiment analysis, data were analyzed in R [*version 4.1.2*, 18] with a binomial generalized linear mixed effects model using the *lme4* package [*version 1.1–27.1*, 19]. The dependent variable was whether the participant produced the TARGET response (coded as a "success") or the COMPETITOR response (coded as a "failure"). Fixed effects comprised the COVID/CONTROL manipulation, the COUGH/NOISE manipulation, and their interaction; all effects were sum-coded, with factor weights set to +/- 0.5. For all models, we started with a maximal random effects structure and followed a preregistered procedure in which random effects were pared to facilitate convergence (see S1 Text section *Method*: *Model-Fitting Strategy* for details). To test hypotheses, contrasts were computed on the fitted model using the *emmeans* package [*version 1.6.0*, 20].

We made two predictions. First, we predicted higher restoration rates to the target word in the COVID vs. CONTROL condition [p(*mask* | #*æsk*) > p(*map* | #*æp*)], indicating increased availability of pandemic-related words compared to pandemic-unrelated words. We tested this hypothesis among trials on which phonemes were replaced by grey NOISE, as this captured increased frequency of pandemic-related words without confound from potentially pandemic-priming coughs. Second, we predicted that restoration of COVID targets (but not CONTROL targets) would increase when replaced by a COUGH relative to grey NOISE, indicating that a coughing sound primed an associative network of pandemic-related words. As all hypotheses were directional, one-tailed hypothesis tests were used as described in the preregistrations, with an alpha level of .05.

## Results

Effect sizes, confidence intervals, and significance tests for each hypothesis test are depicted graphically in Fig 2 and reported numerically in Table 2 for each experiment individually and

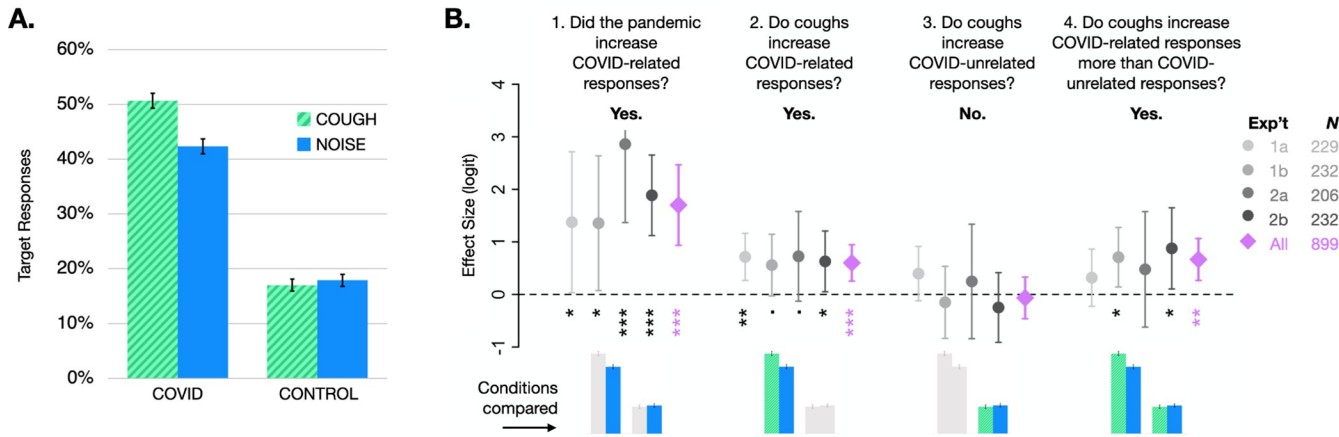

**Fig 2. Results for all experiments.** (A) Cross-experiment (pooled) condition means and standard errors. (B) Effect sizes and 95% confidence intervals for the four comparisons of interest for each experiment (grey circles) and data from all experiments combined (pink diamonds). Colored bars below refer to (A) and show which conditions are involved in each comparison. Asterisks denote significance (***$p < .001$, **$p < .01$, *$p < .05$, and "." $p < .10$).

for all four experiments together. In the text below, all descriptions of results refer to the cross-experiment analysis.

## Preregistered analyses

We found that the pandemic has indeed reorganized lexical knowledge. Across experiments, participants heard ambiguous input as COVID-TARGET words (*mask*) 3.68 times as often as CONTROL-TARGET words (*map*) ($p < .001$; see Fig 2 panel B1). In addition, listeners restored ambiguous input to TARGET words 1.44 times as often in the presence of an interfering COUGH compared to NOISE for COVID word pairs ($p < .001$; panel B2), versus 0.98 times as often for CONTROL word pairs ($p = .539$; panel B3). That is, listeners restored to *mask* more often in the presence of a COUGH compared to NOISE, but listeners restored to *map* equally often in the presence of a COUGH and NOISE, a significant interaction ($p = .002$; panel B4). An identical pattern of statistical significance was obtained when analysis was restricted to the three experiments that were preregistered (Exps. 1b, 2a, 2b; see S1 Text section *Cross-experiment analysis for preregistered experiments only* for details).

## Exploratory analyses

**Alternate explanations.** To determine whether these results were driven by awareness of the purpose of the study rather than unconscious reorganization of the lexicon, we administered a questionnaire after Experiments 1b and 2b asking participants if they thought the experiment was about the pandemic (among other topics). We then analyzed the "unaware" ($n = 290$) and "aware" ($n = 178$) sub-groups separately. Each sub-group's pattern of results and statistical significance was identical to that of the overall analysis, suggesting that task demands did not artificially induce the linguistic effects that were observed (see S1 Text section *Awareness* for details).

We also considered whether the results could be attributable to semantic self-priming. Under this account, after restoring to a COVID-related word, a participant might have been more likely to restore additional COVID-related words on subsequent trials due to (either conscious or unconscious) semantic priming between trials. However, statistical analyses provided no evidence for a self-priming account (see S1 Text section *Self-priming* for details).

**Table 2. Results of all hypothesis tests conducted for individual experiments and across experiments.**

| Question | Comparison | Experiment | *β* | 95% CI | *z* | *p* |
|---|---|---|---|---|---|---|
| 1. Did the pandemic increase COVID-related responses? | COVID/Noise—Control/Noise | 1a | 1.37 | [0.03, 2.71] | 1.69 | **.046** |
| | | 1b | 1.36 | [0.07, 2.64] | 1.74 | **.041** |
| | | 2a | 2.86 | [1.37, 4.35] | 3.15 | **< .001** |
| | | 2b | 1.89 | [1.12, 2.65] | 4.05 | **< .001** |
| | | All | 1.66 | [0.88, 2.44] | 3.49 | **< .001** |
| 2. Do coughs increase COVID-related responses? | COVID/Cough—COVID/Noise | 1a | 0.72 | [0.27, 1.16] | 2.62 | **.004** |
| | | 1b | 0.56 | [-0.03, 1.14] | 1.57 | *.059* |
| | | 2a | 0.73 | [-0.13, 1.58] | 1.40 | *.081* |
| | | 2b | 0.63 | [0.05, 1.21] | 1.79 | **.037** |
| | | All | 0.66 | [0.32, 1.01] | 3.15 | **< .001** |
| 3. Do coughs increase COVID-unrelated responses? | Control/Cough—Control/Noise | 1a | 0.40 | [-0.12, 0.91] | 1.26 | .104 |
| | | 1b | -0.15 | [-0.84, 0.53] | -0.36 | .641 |
| | | 2a | 0.25 | [-0.84, 1.34] | 0.37 | .354 |
| | | 2b | -0.25 | [-0.91, 0.42] | -0.62 | .731 |
| | | All | -0.02 | [-0.42, 0.37] | -0.10 | .539 |
| 4. Do coughs increase COVID-related responses more than COVID-unrelated responses? | (COVID/Cough—COVID/Noise)—(Control/Cough—Control/Noise) | 1a | 0.32 | [-0.22, 0.86] | 0.96 | .167 |
| | | 1b | 0.71 | [0.14, 1.27] | 2.06 | **.020** |
| | | 2a | 0.48 | [-0.62, 1.58] | 0.71 | .237 |
| | | 2b | 0.88 | [0.10, 1.65] | 1.87 | **.031** |
| | | All | 0.69 | [0.29, 1.08] | 2.83 | **.002** |

*Note*. Effect slopes (*β*) represent the change in log-odds ratios between conditions. All predictors were coded so as to yield a numerically positive effect if the answer to the corresponding question was affirmative. As specified in preregistrations, all hypothesis tests were one-tailed. *p*-values are shown in bold for statistically significant effects ($p < .05$) and in *italics* for marginally significant effects ($.05 < p < .10$).

**Word frequency effects.** In the analyses reported above, we created groups of words to use in an experiment with a factorial design. Although we confirmed that the only group of stimuli that significantly increased in word frequency from pre-pandemic (2019) to post-pandemic onset (2020) were the COVID-TARGET words (e.g., *mask*), there was substantial variability within each group. Accordingly, we conducted an analysis across COVID and CONTROL word pairs to determine whether target response rates were related to the relative change in TARGET vs. COMPETITOR word frequency from 2019 to 2020, using the

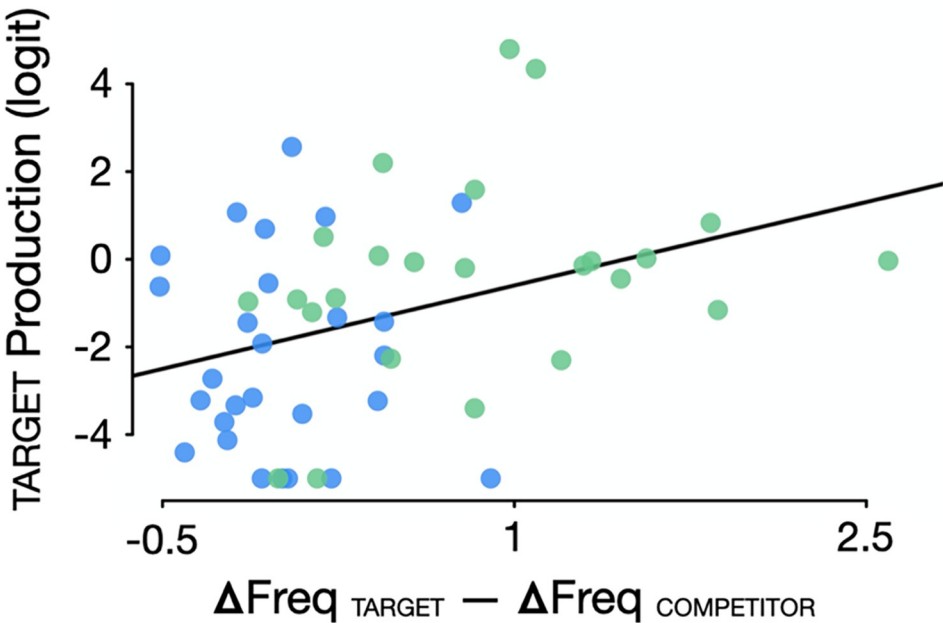

**Fig 3. The more the frequency of a TARGET word (e.g., *mask* or *map*) increased from 2019 to 2020 relative to its COMPETITOR (*task* or *tap*) (x-axis), the more participants perceived the TARGET word relative to its COMPETITOR when replaced by noise (y-axis) ($r(49) = .33$, $p = .016$).** Each dot represents one stimulus: Green dots represent COVID word pairs and blue dots represent CONTROL word pairs.

computed real-world word frequencies of the stimuli [17]. As shown in Fig 3, target response rates were significantly higher for word pairs in which the TARGET increased in frequency more than the COMPETITOR did. That is, words which suddenly became more prevalent in societal discourse as a result of the pandemic were concomitantly more likely to be heard from ambiguous input compared to their minimal pair competitors.

## Discussion

In four experiments, we demonstrated that lexical expectations and comprehension can shift due to rapid changes in a listener's real-life linguistic environment, and that such changes can persist over time. During the COVID-19 pandemic, words such as *mask* and *isolation* suddenly became much more frequent in the global discourse. We postulated that this marked increase in frequency led to an increase in lexical accessibility. This was measured in the current experiments by increased perceptions of pandemic-related words, as compared to pre-pandemic frequency-matched competitors, when faced with an ambiguous auditory input signal.

These results demonstrate that repeated exposure to certain words over a relatively short time span–in comparison to a lifetime's worth of language exposure–leads to dramatic and persistent changes to the language comprehension system. These effects include changes in words' accessibility, as well as the creation of novel conceptual networks (new semantic associations between pandemic-related words and non-linguistic sounds). In addition, we have shown that these effects persist over at least ten months.

Participants were not simply more likely to perceive pandemic-related words overall; the likelihood was linked to each word's individual increase in frequency during the pandemic. This sheds light on an important aspect of the mechanism underlying linguistic adaptation: It indicates that language users weight recent experiences more heavily than older experiences,

not only over the short term–the process which underlies standard semantic and repetition priming effects–but over the long term as well. Furthermore, the only reason we were able to isolate the effects of recent linguistic experience was because of how abruptly the language input changed: The sudden increase in frequency for COVID-TARGET words (e.g., *mask*) in early 2020 made it possible to mathematically dissociate recent and prior linguistic experience.

## Experimental novelty

The present work goes beyond prior research on the effects of lexical exposure in three ways. First, as far as we are aware, this is the first study to use naturally occurring events as the exposure manipulation. Previous research into the effects of word frequency [10, 12] and concept formation [21] has largely relied on artificial manipulations and laboratory settings to induce priming and to provide linguistic exposure to participants. Here, we demonstrate that comprehenders can quickly and flexibly adapt their expectations to accommodate changes in the statistical properties of their linguistic environment. By bringing this finding from the laboratory into the real world, our work follows in the footsteps of recent research using world events to study other psycholinguistic phenomena [22].

Second, the results of these experiments demonstrate the presence of *long-lasting* changes in lexical accessibility induced by rapid changes in the linguistic input. Prior work has demonstrated that such accessibility changes can be induced rapidly, after just a few minutes or exposure tokens [e.g., 11], but demonstrations of so-called "long-term" effects have been confined to periods of less than one day. In contrast, the present work demonstrates lasting effects on language comprehension over the course of nearly a year, at least insofar as the stimulus (i.e., the heightened frequency of pandemic-related words) remains present. This observation has important consequences for understanding how language users weigh their recent linguistic experience against statistical expectations built up over a lifetime of language exposure, suggesting that the former can outweigh the latter.

Finally, our experiment removes both the homogenous experimental context and the temporal proximity between prime and target presentation. Most prior research on priming and lexical access occurs within a single experimental context: The participant is in the lab, sitting in a testing room, hearing primes and targets from the same computer speakers, and so forth. Memory tends to be enhanced by such contextual consistency [23], as demonstrated by extreme manipulations such as divers learning a word list either on land or underwater, and then demonstrating better recall in the same environment [24]. Our experiments, however, had no such contextual overlap: The priming "phase" occurred wherever–and whenever–participants interacted with COVID-19-related words when going about their lives. In spite of these inconsistencies between prime and target experience, we nonetheless observed strong and consistent effects across experiments, indicating that the linguistic adaptation we report here is robust to real-world situations.

## Future directions

Several factors may limit the generalizability of our results. First, we tested only a very restricted set of words, a tiny fraction of those that the average adult speaker knows. It remains an open question whether and how the full lexicon is affected by changes in frequency of a small number of words. Second, the demonstrated effects occurred as the result of a sudden and enormous shift in frequency of the critical items. Most real-world lexical experience consists of substantially more modest and gradual changes in a word's usage. It is possible that the learning and access processes which underlie these more gradual effects are governed by different mechanisms than those captured in the present work. Both of these limitations were by

design, taking advantage of the exigencies of the COVID-19 pandemic to demonstrate an extreme case of change in lexical frequency, accessibility, and changes to semantic networks. An important question for future research will be whether such effects can be demonstrated on a smaller scale when the concomitant lexical changes are milder as well.

Another question for future research, if discussion of the pandemic ever recedes and the frequency of words like *mask* returns to 2019 levels, is whether the increased accessibility observed in the present experiments remains high for these words, and for how long after their frequency decreases. This is an important open question which would elucidate the time course of changes to the comprehension system on the basis of changing input.

## Conclusions

The COVID-19 pandemic precipitated a massive change in the frequency of certain words. By leveraging this change in a natural experiment, we have demonstrated that sudden changes in recent linguistic input can have measurable and lasting impacts on how listeners process speech, suggesting that lexical comprehension can be affected for the long-term by abrupt and short-term changes.

## Supporting information

**S1 Text. Supplemental method and results.**
(DOCX)

## Acknowledgments

The authors thank Riley Adachi and Mohit Gurumukhani for assistance with stimulus creation, data collection, and response coding.

## Author Contributions

**Conceptualization:** Daniel Kleinman, Adam M. Morgan, Rachel Ostrand, Eva Wittenberg.

**Data curation:** Daniel Kleinman.

**Formal analysis:** Daniel Kleinman, Adam M. Morgan, Rachel Ostrand.

**Funding acquisition:** Rachel Ostrand, Eva Wittenberg.

**Investigation:** Eva Wittenberg.

**Methodology:** Daniel Kleinman, Adam M. Morgan, Rachel Ostrand, Eva Wittenberg.

**Resources:** Rachel Ostrand.

**Visualization:** Daniel Kleinman, Adam M. Morgan.

**Writing – original draft:** Daniel Kleinman, Adam M. Morgan, Rachel Ostrand, Eva Wittenberg.

**Writing – review & editing:** Daniel Kleinman, Adam M. Morgan, Rachel Ostrand, Eva Wittenberg.

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
