## [Decision Letter · Decision Letter 0]

11 May 2022

PONE-D-22-09945Lasting Effects of the COVID-19 Pandemic on Language ProcessingPLOS ONE

Dear Dr. Ostrand,

Thank you for submitting your manuscript to PLOS ONE. After careful consideration, we feel that it has merit but does not fully meet PLOS ONE’s publication criteria as it currently stands. Therefore, we invite you to submit a revised version of the manuscript that addresses the points raised during the review process.

Congratulations on a great paper!! As you will see, one of the reviewers has recommended that we accept the manuscript as is and the other reviewer only has very minor comments. I ask that you address those as you see fit. I will not send out your revised manuscript for another round of reviews.

We look forward to receiving your revised manuscript.

Kind regards,

Stephanie Ries-Cornou, Ph.D

Academic Editor

PLOS ONE

Journal Requirements:

2. Please note that in order to use the direct billing option the corresponding author must be affiliated with the chosen institute. Please either amend your manuscript to change the affiliation or corresponding author, or email us at plosone@plos.org with a request to remove this option.

Reviewers' comments:

Reviewer's Responses to Questions

**Comments to the Author**

1. Is the manuscript technically sound, and do the data support the conclusions?

Reviewer #1: Yes

Reviewer #2: Yes

2. Has the statistical analysis been performed appropriately and rigorously? 

Reviewer #1: Yes

Reviewer #2: Yes

3. Have the authors made all data underlying the findings in their manuscript fully available?

Reviewer #1: Yes

Reviewer #2: Yes

4. Is the manuscript presented in an intelligible fashion and written in standard English?

Reviewer #1: Yes

Reviewer #2: Yes

5. Review Comments to the Author

Reviewer #1: The manuscript reports a clever set of experiments testing whether the change in frequency for pandemic related words changed their processing. Using a phonemic restoration task, the results showed a bias to hear ambiguous stimuli as Covid related words rather than pre-Covid frequency matched control words. In addition, using a cough instead of grey noise for the word interruption further increased the bias for Covid related words. The results demonstrated processing consequences of recent frequency increases and changes in semantic priming patterns.

The results are unsurprising. However, the circumstances for collecting the data are novel and the results confirm that one sees these expected effects under these novel conditions.The comparison of a cough intrusion with a noise one to test for priming is inspired. The methodology and analyses are solid. The writing is unusually clear and concise. I strongly recommend this manuscript for publication. No changes needed.

Reviewer #2: First off, this paper was generally very easy to read!

Next, some thoughts from a phonetician: fricatives and voiceless stops have real noise components, and I would have expected that this would make listeners more likely to restore towards "task" than towards "mask." I'm not intimately familiar enough with the phoneme restoration literature to know whether this effect has been documented or not. Either way, this is of course not a confound for your effect in cases like mask/task, quite the opposite, and strengthens the case that a phonetic phenomenon is not responsible for the higher proportion of COVID target restorations. Just something to think about and comment on, if you wanted to.

I appreciate that all the data and materials are available at the additional link, but I think it would be nice to have a clearer explanation of how experiments 1 and 2 differed from each other in the text, and also to have a stimuli table.

I'd like to know how many minutes the study typically lasted if participants were only paid $2.50. It's important to be transparent about how we treat participants.

6. PLOS authors have the option to publish the peer review history of their article (what does this mean?). If published, this will include your full peer review and any attached files.

Reviewer #1: No

Reviewer #2: No

---

## [Author Response · Author response to Decision Letter 0]

16 May 2022

Next, some thoughts from a phonetician: fricatives and voiceless stops have real noise components, and I would have expected that this would make listeners more likely to restore towards "task" than towards "mask." I'm not intimately familiar enough with the phoneme restoration literature to know whether this effect has been documented or not. Either way, this is of course not a confound for your effect in cases like mask/task, quite the opposite, and strengthens the case that a phonetic phenomenon is not responsible for the higher proportion of COVID target restorations. Just something to think about and comment on, if you wanted to.

We added a comment that our design obviates any concerns about different phonemic/phonetic properties between different types of consonants, as they are matched in the COVID and CONTROL conditions (pg. 9-10):

“…this ensured that any differences between phonemes that might affect restoration (e.g., the presence of noise components in fricatives and voiceless stops) would affect the COVID and CONTROL conditions equally.”

I appreciate that all the data and materials are available at the additional link, but I think it would be nice to have a clearer explanation of how experiments 1 and 2 differed from each other in the text, and also to have a stimuli table.

We have added a comment about the differences between Exps 1 and 2 on page 9:

“The only experimental design difference between Experiments 1 and 2 was the specific stimuli used.”

We have also added a stimuli table (Table 1, page 12).

I'd like to know how many minutes the study typically lasted if participants were only paid $2.50. It's important to be transparent about how we treat participants.

We have added a note about the length of the study (10 minutes) on page 7.

---

## [Editor Report · Decision Letter 1]

18 May 2022

Lasting Effects of the COVID-19 Pandemic on Language Processing

PONE-D-22-09945R1

Dear Dr. Ostrand,

We’re pleased to inform you that your manuscript has been judged scientifically suitable for publication and will be formally accepted for publication once it meets all outstanding technical requirements.

Kind regards,

Stephanie Ries-Cornou, Ph.D

Academic Editor

PLOS ONE
---

## [Editor Report · Acceptance letter]

25 May 2022

PONE-D-22-09945R1 

Lasting Effects of the COVID-19 Pandemic on Language Processing 

Dear Dr. Ostrand:

I'm pleased to inform you that your manuscript has been deemed suitable for publication in PLOS ONE. Congratulations! Your manuscript is now with our production department. 

Kind regards, 

on behalf of

Dr. Stephanie Ries-Cornou 

Academic Editor

PLOS ONE